# A Cross-Sectional Study of the Relationship of Timed Up & Go Test with Physical Characteristics and Physical Activity in Healthy Japanese: The Toon Health Study

**DOI:** 10.3390/healthcare9080933

**Published:** 2021-07-24

**Authors:** Yuichi Uesugi, Koutatsu Maruyama, Isao Saito, Kiyohide Tomooka, Yasunori Takata, Ryoichi Kawamura, Haruhiko Osawa, Takeshi Tanigawa, Yoshihiko Naito

**Affiliations:** 1Department of Food Sciences and Nutrition, School of Food Sciences and Nutrition, Mukogawa Women’s University, Nishinomiya 663-8558, Japan; naito@mukogawa-u.ac.jp; 2Department of Bioscience, Graduate School of Agriculture, Ehime University, Matsuyama 790-8566, Japan; maruyama.kotatsu.rt@ehime-u.ac.jp; 3Department of Public Health and Epidemiology, Faculty of Medicine, Oita University, Yufu 879-5593, Japan; saitoi@oita-u.ac.jp; 4Department of Public Health, Graduate School of Medicine, Juntendo University, Tokyo 113-8421, Japan; k-tomoka@juntendo.ac.jp (K.T.); tataniga@juntendo.ac.jp (T.T.); 5Department of Diabetes and Molecular Genetics, Graduate School of Medicine, Ehime University, Toon 791-0295, Japan; ytakata@m.ehime-u.ac.jp (Y.T.); kawamura.ryoichi.me@ehime-u.ac.jp (R.K.); harosawa@m.ehime-u.ac.jp (H.O.)

**Keywords:** physical function, motor function, amount of exercise, walking speed, frail, sarcopenia, Japan

## Abstract

This study evaluated the Timed Up & Go test (TUG) among healthy Japanese individuals without walking problems to clarify the relationship of TUG performance with physical characteristics and physical activity according to sex and age groups. In total, 797 men and women (30–84 years old) in Toon City, Ehime Prefecture, were assessed from 2016 to 2017. The survey data for physical characteristics, TUG performance, and physical activity measures were used. After adjusting for age according to TUG time and categorization into sex and age groups (30–64 and 65–84 years), the relationship of TUG performance with physical characteristics and physical activities was confirmed using multiple regression analysis. In men, TUG performance was associated with work and household chores in the 30–64-year age group, whereas it was only associated with skeletal muscle mass among those older than 65 years. In women, TUG performance was associated with height and amount of exercise, regardless of age. In conclusion, TUG performance may be maintained by increasing the amount of physical activity for men through work and housework, and increasing the amount of exercise for women, which may prevent the need for long-term care in the future.

## 1. Introduction

The proportion of the aged in the Japanese population is estimated to increase to 28.4% in 2020 and to further increase to 38.4% by 2065 [1]. Additionally, the number of people certified for long-term care is increasing yearly, and was estimated at 6,282,000, with the elderly aged ≥ 75 years accounting for 31.9% of this population in 2017 [1]. In this scenario, it is clear that the number of people certified for long-term care will increase further, which would add pressure on the budgets of national and local governments [2]. Therefore, to solve the problem of increasing healthcare costs due to long-term care requirements, it is necessary to curb the abovementioned increase and prolong health expectancy. We believe that preventive measures are necessary not only for the elderly but also for healthy adults to ensure that individuals do not develop a disability that necessitates long-term care [3,4].

Maintaining physical functions, such as muscle strength, is important for developing a healthy expected quality of life. Walking speed, 6-min walking test, locomotive syndrome risk test, and Timed Up & Go test (TUG) are used to measure physical function [5,6,7,8,9]. These measures assess the physical functions required for performing the activities of daily living, mainly among the elderly. Among these, the TUG was created to clinically assess the dynamic balance of the elderly by Podsiadlo and Richardson in 1991 [10] and has been gaining ground, in recent years, as a test that can easily evaluate walking ability, flexibility, and agility [10]. This test has been used not only to assess the risk of falls in the elderly due to sarcopenia, but also to evaluate physical function in previous studies with participants who had cerebral palsy, stroke, and Parkinson’s disease [11,12,13]. In addition, some studies were conducted with a focus on children and adolescents [13,14], and another study has shown that TUG performance is associated with the level of physical activity [15]. Therefore, we believe that TUG is useful for evaluating the physical function of healthy adults.

Very few studies using TUG have been conducted on healthy adults. In addition, despite the relationship between TUG performance and physical activity, no study has confirmed the quality and quantity of physical activity that is required. Other studies have shown differences in physical activity performance by sex and age, and these differences may be related to TUG performance [16,17]. If the association between TUG performance and physical activity performance is clarified, preventive measures to diminish the requirement for long-term care can be implemented. Therefore, we conducted the TUG in a cohort of healthy Japanese individuals who have almost no walking problems to clarify the relationship of TUG performance with physical characteristics and physical activity as well as the differences in this relationship according to sex and age groups.

## 2. Materials and Methods

### 2.1. Study Population and Design

This cross-sectional study was conducted as part of the Toon Health Study [18]. The participants of this study were 811 men and women aged 30–84 years living in Toon City, Ehime, between 2016 and 2017. Toon City has a population of approximately 33,000, of which nearly 23,000 are between the ages of 30 and 84. The participants of this study represent 3.5% of the population of this city. The participants were enrolled voluntarily after being informed of the purpose and requirements of the study. We provided this information through the media in newspapers or via posters and advertisements on the internet. Furthermore, voluntary health groups contacted their members to invite them to participate in the study. Only certified residents aged over 30 years were enrolled in the study, and pregnant women or individuals with an illness that was considered severe by a physician were excluded from this study. Of the enrolled participants, we selected those who had undergone a TUG and who were not certified for long-term care. Using these criteria, 797 individuals (285 men and 512 women) were included in the analysis. Written informed consent was obtained from all participants.

### 2.2. Measurement and Measuring Equipment

#### 2.2.1. Timed Up & Go Test

The participants were observed and timed while they rose from an armchair, walked 3 m, turned, walked back, and sat down again [10]. We paid close attention to participants who fell. The measurements were performed twice, and the average time (in seconds) was recorded for analysis.

#### 2.2.2. Questionnaire Survey

The questionnaire survey was assessed using the Japan Arteriosclerosis Longitudinal Study Physical Activity Questionnaire (JALSPAQ) [19,20], and the accuracy was validated using the double-labeled water method [19]. We asked participants about the frequency of daily and weekly sleep, job, transportation, housework, exercise and leisure activities (e.g., gardening, home carpentry, recreation or voluntary work). The participants reported the frequency and duration of physical activity in each domain. They were also asked about the approximate percentage of time sitting, standing, and walking in the job domain. Additionally, they were asked about the type, frequency per month, and duration of exercise and leisure activity. The responses for each physical activity were converted to metabolic equivalents (METs) according to the Compendium of Ainsworth et al. and expressed as METs-h/day. Furthermore, the time that the participants expended in these activities was added and then subtracted from 24 h. This resulting difference was defined as the time that was spent in sedentary behaviors. Moreover, we estimated the physical activity in each domain as well as the total physical activity and exercise amount per day using dedicated software.

#### 2.2.3. Physical Characteristics

Physical characteristics included height, body weight, and body composition. Body weight and body composition were measured using a portable MC-780A scale (Tanita Corporation, Tokyo, Japan) equipped with bioelectrical impedance analysis technology. Its accuracy was validated using dual-energy X-ray absorptiometry [21]. Body composition was estimated based on skeletal muscle mass, body fat mass, and body fat percentage using the bioelectrical impedance method. The body mass index (BMI) was calculated by dividing the body weight (kg) by the square of the height (m).

### 2.3. Statistical Analysis

Pearson’s correlation coefficient between TUG performance and age was calculated. We compared TUG time between sexes with and without stratification by age group (30–64 years and 65 years and older) using the same test. We then conducted a multiple regression analysis using forward-backward stepwise selection method for each sex and age group with age-adjusted TUG time as the objective variable, and four physical characteristics (height, body weight, body fat mass, and skeletal muscle mass) and seven physical activities as independent variables. IBM SPSS Statistics ver. 22.0 for Windows (IBM Corp., Armonk, NY, USA) was used in this statistical analysis, and the significance level was set at 5% (two-sided test).

### 2.4. Ethical Considerations

This study was approved by the Institutional Review Board of Ehime University Hospital (authorization number: 1705011) and Mukogawa Women’s University (authorization number: 20–116). The study was conducted in accordance with the Code of Ethics of the World Medical Association (Declaration of Helsinki) for experiments involving humans.

## 3. Results

### 3.1. Characteristics of the Participants

Table 1 shows the age, physical characteristics, and physical activities by sex among the 797 participants.

### 3.2. Correlation between TUG and Age

Figure 1 shows the correlation between subject age and TUG time. We observed a weak but positive correlation (r = 0.326, *p* < 0.001).

The dispersion in TUG time increased as the participants grew older. Five participants aged ≥60 years exceeded the cut-off value of 11 s for the musculoskeletal ambulation disability symptom complex set by the Japanese Orthopedic Association. One of these participants exceeded the cut-off value of 13.5 s for the risk of falls.

### 3.3. Comparison of TUG Time by Sex and Age Group

Table 2 shows the results of the comparison of TUG time by sex and the presence or absence of old age. The women’s mean time was significantly higher than that of the men. The mean time for the >65-year age group was significantly higher than that of the 30–64-year age group among both men and women.

### 3.4. Stepwise Multiple Regression Method of Factors Related to TUG Time by Sex and Age Group

Table 3 shows the results of stepwise multiple regression analysis. In men aged 30–64 years, TUG performance was associated with three physical activities, whereas in men over 65 years of age, it was associated with skeletal muscle mass. In women aged 30–64 years, TUG performance was associated with height, leisure, and exercise, whereas in women who were older than 65 years, TUG performance was associated with height, sleep, and exercise.

## 4. Discussion

This study aimed to conduct TUG among healthy Japanese individuals without walking problems, to clarify the relationship of TUG performance with physical characteristics and physical activity, by sex and age group. 

We confirmed that TUG performance became slower as the subjects got older, regardless of sex. In addition, we confirmed that the dispersion in TUG times tended to increase as age increased. This result was similar to that of a previous study of people in their sixties to eighties [22] and that of a study on walking speed [23]. Furthermore, five subjects in their late sixties exceeded the cut-off value of 11 s for the musculoskeletal ambulation disability symptom complex and the cut-off of 13.5 s for fall risk [24,25]. In contrast, some of the participants in their seventies had the same performance time as that of participants in their thirties. Therefore, these participants were considered to have a low musculoskeletal decline with age. 

Comparing TUG times by sex and age group, men took significantly less time to complete the test than women. For both men and women, the TUG time of those over 65 years old was significantly higher than that of those aged 30–64 years. A decline in physical performance with aging, such as muscle strength, and men having better physical performance than women, are known facts [26,27]. These results are the same as those reported in previous studies [22]. The authors linked these results to similar age-related decline rates in physical performance, namely, gait, balance, and hand grip strength in both sexes [28]. Other studies have shown that the results of the locomotive syndrome risk test were also affected by sex and age [29]. Therefore, age and sex should be considered when assessing adults using TUG. 

We used multiple regression analysis categorized by sex and age group, to investigate factors related to TUG, with physical conditions and physical activities as independent variables. First, the TUG time for men aged 30–64 years was found to be related with job activity, leisure physical activity, and housework. Job activity was negatively associated with TUG. This result suggested that low physical activity in a job decreases the TUG performance. In Japan, the sitting duration at work is longer than that reported in other countries [30]. Long sitting durations lead to deterioration of posture and physical ability, such as in muscle strength [31,32]. Therefore, especially in Japan, it is necessary to introduce standing meetings, desk work, and a time and place for physical activity in the office to reduce the sitting time. Second, there was a positive association between leisure-time physical activity and TUG time. This result was similar for women aged 30–64 years. Leisure is distinguished from exercise and indicates the amount of light physical activity in the JALSPAQ. This includes sedentary hobbies, such as reading, driving, and performing music. Furthermore, high physical activity in leisure indicates that individuals spend a lot of time on those hobbies, which means less time for work, housework, and exercise. Therefore, individuals aged 30–64 years with high leisure time may have a shorter time for other activities, and this may have contributed to the association with TUG. Third, there was a negative association between housework physical activity and TUG time. The results show that housework chores improve TUG performance. The percentage of Japanese men performing household chores is much lower than that of foreigners and Japanese women, and many Japanese men do not carry out chores at home [33]. Performing housework is associated with inactivity [34], and increased inactivity time has been shown to lead to frailty [35]. Therefore, it can be said that it is important for men to take on roles such as household chores and increase physical activity to maintain TUG performance and prevent frailty.

There was a negative association between skeletal muscle mass and TUG time in men aged ≥ 65 years. The results support that TUG can be used to measure muscle mass and movement disorders, such as weakness and instability. We believe that a factor that is relevant only to men is that Japanese men lose more muscle mass (especially lower limb muscles) with age than women [36]. Therefore, it is considered that measures against muscle mass loss from a young age may lead to the maintenance of TUG performance in men. However, there was no association between skeletal muscle mass and TUG performance in participants aged 30–64 years. Studies examining the relationship between locomotive syndrome and gait and body composition have shown that body composition does not affect the normal walking speed in adolescents [37]. The same association applies to TUG, and if there is a specific amount of skeletal muscle mass, then exercise at the level of daily living does not affect TUG performance; therefore, this is considered to indicate that no association was found. However, regardless of the sex, loss of skeletal muscle mass leads to frailty and interferes with daily life; therefore, it is important to suppress the loss of skeletal muscle mass.

In women, there was a negative association of height and exercise with TUG, regardless of age group. This result suggests that taller individuals and those with high physical activity during exercise have improved TUG performance. Previous studies have shown that age and sex are associated with TUG, but the association with height is a new finding [22]. Studies on walking speed have shown that height is associated with walking speed [38,39]. In the locomotive syndrome test, the result of dividing a maximum of two steps by the height was used as a criterion for locomotive syndrome [9]. TUG performance is also thought to be height-dependent, as TUG time for women in this study was not associated with other physical indicators. Therefore, when using the TUG, it is necessary to set a cut-off value that considers height. However, there was no association between TUG and height in men, suggesting that many healthy participants should be considered.

This study has some limitations. First, as mentioned above, this study is a cross-sectional study, so it is not possible to mention the causal relationship. Second, because our subjects were volunteers from a single community and the disproportion in the number of male and female participants contributes to a possible bias; therefore, our findings may not be representative of the general Japanese population. Third, since the impedance method and the questionnaire survey were used, although the validity has been proven, there may be an error from the actual value. Despite the abovementioned limitations, the study found a close relationship between TUG performance and physical composition and physical activity in healthy participants with normal gait function. 

## 5. Conclusions

This study evaluated TUG performance among healthy Japanese individuals without walking problems to clarify the relationship of TUG performance with physical characteristics and physical activity, by sex and age groups. We found that TUG performance in men aged 30–64 years was associated with work, leisure, and housework time, whereas in men aged 65 years and older, the TUG performance was associated with skeletal muscle mass. However, in women, there was an association between height and exercise regardless of age. Therefore, we found that the TUG performance is associated with physical characteristics and physical activity, even in healthy individuals. We recommend that men should increase their physical activity, such as in-job activity and housework and that women should exercise to control the decrease in skeletal muscle mass with age to prevent the need for future long-term care.

## Figures and Tables

**Figure 1 healthcare-09-00933-f001:**
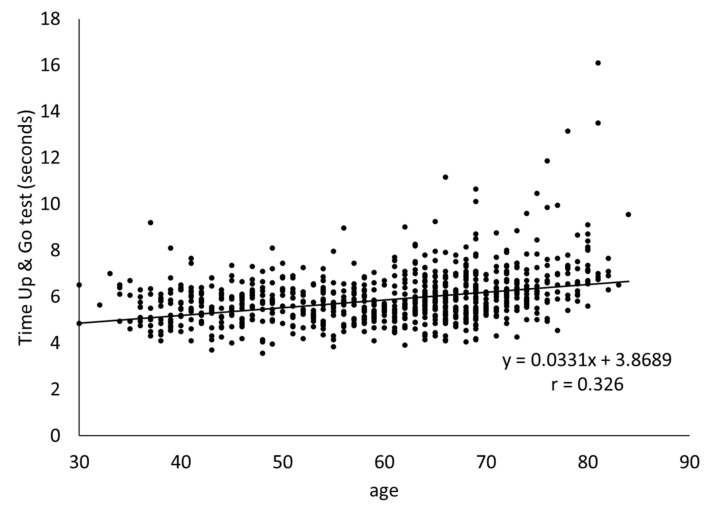
Relationship between age and result of Timed Up & Go test.

**Table 1 healthcare-09-00933-t001:** Physical characteristics ^†^.

		Men (*n* = 285)	Women (*n* = 512)
Age	(years)	61.4 ± 12.6	58.9 ± 11.8
**Body index**			
Height	(cm)	167.0 ± 0.4	154.6 ± 0.3
Body weight	(kg)	66.7 ± 0.6	53.6 ± 0.4
Body weight index	(kg/m^2^)	23.9 ± 0.2	22.4 ± 0.2
Skeletal muscle mass	(kg)	52.5 ± 0.4	37.3 ± 0.2
Body fat mass	(kg)	14.1 ± 0.3	16.3 ± 0.3
Body fat percentage	(%)	20.7 ± 0.3	29.5 ± 0.3
**Physical activity**			
Sleep	(h)	7.2 ± 1.2	6.9 ± 1.1
Job	(Mets∙h)	8.6 ± 10.8	5.1 ± 5.9
Transportation	(Mets∙h)	1.7 ± 1.7	1.7 ± 1.5
Housework	(Mets∙h)	1.7 ± 2.3	8.3 ± 4.4
Exercise	(Mets∙h)	1.6 ± 2.4	1.0 ± 1.3
Leisure	(Mets∙h)	1.0 ± 3.4	0.5 ± 1.2
Sedentary	(Mets∙h)	13.0 ± 5.9	12.3 ± 3.2
Total	(Mets∙h)	34.8 ± 5.6	35.7 ± 3.8

^†^ Data presented as mean ± standard deviation.

**Table 2 healthcare-09-00933-t002:** Mean difference in TUG time by sex and age group.

				TUG Time (Seconds)	
		*n*	(%)	Mean ± SD	*p*-Value ^†^
**Sex**	Men	285	35.8	5.9 ± 1.2	0.043
	Women	512	64.2	6.1 ± 1.2	
**Age group**	**Men**				
	30–64	141	49.5	5.5 ± 1.0	<0.001
	≥65	144	50.5	6.2 ± 1.2	
	**Women**				
	30–64	317	61.9	5.8 ± 0.8	<0.001
	≥65	195	38.1	6.5 ± 1.5	

TUG = Timed Up & Go; SD = standard deviation. ^†^ Independent *t*-test.

**Table 3 healthcare-09-00933-t003:** Stepwise multiple regression analysis of factors related to TUG time by sex and age group ^†^.

	Variable	B	SE	β	R^2^
**Men**	**30–64 years**				
	Job	−0.013	0.003	−0.284 **	0.433
	Leisure	0.048	0.015	0.248 **	
	Housework	−0.023	0.010	−0.173 *	
	**≥65 years**				
	Skeletal muscle mass	−0.011	0.002	−0.354 **	0.354
**Women**	**30–64 years**				
	Height	−0.011	0.003	−0.208 **	0.309
	Leisure	0.049	0.017	0.155 **	
	Exercise	−0.032	0.014	−0.122 **	
	**≥65 years**				
	Height	−0.005	0.002	−0.200 **	0.309
	Sleep	0.020	0.008	0.163 **	
	Exercise	−0.004	0.002	−0.144 **	

B = partial regression coefficient; SE = standard error of the mean; β = standardized partial regression coefficient; R^2^ = coefficient of determination. * *p* < 0.01, ** *p* < 0.05. ^†^ The objective variable was TUG time adjusted by age, and the independent variables were four physical characteristics (height, body weight, body fat mass, and skeletal muscle mass) and seven physical activities. When B and β are negative, this shows a decrease in TUG time (improvement in TUG performance) with the increase of each independent variable.

## Data Availability

The data are not publicly available due to privacy restrictions. These data are accessible only to researchers belonging to the Toon Health Study.

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
