# Peer review of "A Cross-Sectional Study of the Relationship of Timed Up & Go Test with Physical Characteristics and Physical Activity in Healthy Japanese: The Toon Health Study"

_healthcare, 2021, doi:10.3390/healthcare9080933_

Round 1

Reviewer 1 Report

Dear Authors,

Thank you for this interesting manuscript. Below are my suggestions and comments.

Lines 95-96: it is more common to define ratio weight, kg/height, m2 as body mass index (BMI).

Please provide the name of the statistical software used for analysis (section 2.3).

Please provide measurement units for physical activities and explain in a detailed way, how they were calculated (section 2.2.2). Also, intensities of these activities should be provided. Why walking was combined with cycling (Table 1)? Job (is it an analogue of occupational physical activity?) - it can be manual with high energy expenditures and non-manual, performed in a sitting position with low energy expenditures. Moreover, in Table 3 job is negatively associated with TUG time makes it even more confusing and unclear - more MET-hours spent in the job produces shorter TUG time and better physical performance (Table 3, men, age group 30-64 years)? What about people who were unemployed and/or retired? Were they excluded from this analysis or did they collect 0 MET-hours from job activities? Also, in younger women, exercise activity seems to prolong TUG time, while in older women - to reduce? How can it be explained? Or probably this is a typo, as in a younger women group B for exercise is negative, while ß is positive. Same for the sleep effect in older women group - B is negative, while ß is positive. Also, it would be interesting to see the effects of all the variables included in regression analyses, not only significant.

Also, please clarify, why did you decide to include height and weight in regression analysis, not body mass index? Did you check for multicollinearity effects?

Description of the regression findings is very scarce. Please add more information explaining the direction of associations between independent variables and TUG time.

What do the p-values (*, **) for the R2 represent? Or is it F statistics?

Please add the main findings to the conclusions section.

Author Response

Dear Reviewer 1,

We are grateful for your prompt and courteous response. Many points were pointed out, and this feedback is very helpful. Thank you for your constructive comments. We have carefully revised our manuscript as suggested. The page and line numbers in the responses refer to the relevant parts of the main manuscript where revisions have been undertaken. We have provided point-by-point responses to each comment. Please check the attached document.

Reviewer 2 Report

The text presented is well structured and organized, but its quality would improve if the following aspects were added:

1.- Clearly explain the criteria used for the selection of the population under study.

2.- Explain the disproportion between men and women.

3.- add the characteristics of the population under study.

4.- Expand the theoretical justification on the subject by adding its importance for the scientific community.

5.- Add the limits of the research and possible publications of these findings.

6.- Add the objectives and hypotheses of the research

Author Response

Dear Reviewer 2,

We are grateful for your prompt and courteous response. Thank you for your constructive comments. We have carefully revised our manuscript as suggested. The page and line numbers in the responses refer to the relevant parts of the main manuscript where revisions have been undertaken. We have provided point-by-point responses to each comment. Please check the attached document.

Reviewer 3 Report

This study evaluated the Timed Up & Go test (TUG) among healthy Japanese individuals without walking problems and to clarify the relationship of TUG performance with 20 physical characteristics and physical activity according to sex. Here are some comments:

  1. P1, line 44: ‘We believe that these measures are necessary not only for the elderly but also for healthy 44 adults’, what were ‘these measures’? The detail measures or variables can be presented.
  2. There were two suggestions about introduction. First, why TUG was used, rather than other physical functions? Second, what were physical characteristics? The definition or variables of physical characteristics can be provided in introduction.
  3. Study population: the population of the Toon health study can be provided, for example how many people aged 30-84 years old in Toon, and then 811 joint the study. In addition, the characteristics of Toon can be provided.
  4. 2.2. Questionnaire Survey: which physical activity scale was used, and the information about item number, scoring, and reliability and validity should be given.
  5. In statistical analysis, TUG was taken as a dependent variable, and physical characteristics and physical activity were independent variables. Therefore, in the introduction, the reasons why physical characteristics and physical activity had potential effects on TUG can be given.
  6. P4, line 139: please add the results of women aged over 65 years.
  7. P6, line 175-179: there was negative association between job activity and TUG, which the participants with more job activity had better TUG performance. However, the author stated ‘sitting time during work is longer in Japan than other countries’ and ‘low physical activity during work causes a decrease in TUG performance’. it was confused for readers and modification will be needed.
  8. P6, line 175-191: more job and housework negatively related to TUG time, but more leisure activity positively related to TUG time. The reasons and potential explanation can be addressed especially the positive relationship between leisure activity and TUG time.
  9. The physical characteristics were only related to TUG in men over 65 years old. The reasons why body composition/skeletal muscle mass was not related to TUG in women and men aged 30-64 years can be discussed.
  10. P6, line 221: the authors stated: ‘women exercise to control body fat mass’, however, body fat mass was not significant in the result.

Author Response

Dear Reviewer 3,

Thank you for your prompt and courteous response. Thank you for pointing out a problem that we did not notice. We have revised the manuscript in accordance with the valuable opinion of the reviewer. The page and line numbers in the responses refer to the relevant parts of the main manuscript where revisions have been undertaken. We have provided point-by-point responses to each comment. Please check the attached document.

Round 2

Reviewer 1 Report

Dear Authors,

Thank you for the improved version of the manuscript, good job!